# Neonatal Morbidity after Cervical Ripening with a Singleton Fetus in a Breech Presentation at Term

**DOI:** 10.3390/jcm11237118

**Published:** 2022-11-30

**Authors:** Laura Berthommier, Lucie Planche, Guillaume Ducarme

**Affiliations:** 1Department of Obstetrics and Gynecology, Centre Hospitalier Departemental, 85000 La Roche sur Yon, France; 2Clinical Research Center, Centre Hospitalier Departemental, 85000 La Roche sur Yon, France

**Keywords:** breech, mode of labor, induction of labor, cervical ripening, neonatal morbidity, maternal morbidity

## Abstract

Vaginal delivery in women with a breech presentation is part of common practice in France despite much debate, and the induction of labor (IOL) with a fetus in a breech presentation at term remains uncommon. Little is known about the effectiveness of cervical ripening and its neonatal and maternal safety in these women. We present a retrospective study of 362 women who gave birth to a live singleton fetus in a breech presentation at term. The objective was to compare severe maternal and neonatal morbidity according to the planned mode of labor (spontaneous labor or the induction of labor (IOL) with a favorable cervix, cervical ripening, or elective cesarean delivery) and, specifically, to compare cervical ripening to the other modes of labor. The rate of severe neonatal morbidity was 3.0% and was significantly higher after the IOL compared to elective cesarean delivery (*p* = 0.02), and the severe maternal morbidity rates were similar. Multivariable logistic regression analysis found no significant association between cervical ripening and either composite severe neonatal (adjusted odds ratio [aOR] 2.80, 95% confidence interval [CI] 0.10–43.6) or maternal morbidity (aOR 1.29, 95% CI 0.05–11.5). Our results support a policy of offering cervical ripening to the appropriately selected candidates with a singleton fetus in a breech presentation at term without increasing the incidence of severe maternal and neonatal morbidity.

## 1. Introduction

Vaginal delivery in women with a breech presentation is part of common practice in France despite much debate [1,2,3], and the induction of labor (IOL) with a fetus in a breech presentation at term remains uncommon [4]. French guidelines indicated that the IOL did not seem to be associated with an increase in perinatal morbidity compared to spontaneous labor and planned cesarean delivery, even in cases of labor induction with an unfavorable cervix that required cervical ripening [5]. Few studies reported an IOL with a breech presentation at term [6,7], and almost all of them did not distinguish between the IOL with a favorable or an unfavorable cervix. We therefore have few specific data on cervical ripening at term in women with a breech presentation. Burgos et al. [6] retrospectively reported no significant difference in neonatal morbidity between the IOL and spontaneous labor among women with breech presentation, IOL, and an unfavorable cervix. In the PREMODA study [3], 76 women (out of a total of 163 IOL) were induced with an unfavorable cervix, and neonatal morbidity was similar according to the mode of labor induction (oxytocin, prostaglandins, or cervical ripening balloon (CRB)). Regarding this large prospective observational study, French guidelines indicated that cervical ripening and vaginal prostaglandins seemed possible in women with a breech presentation, but there was no recommendation about CRB or misoprostol [5]. The two main questions were the effectiveness of these practices and their neonatal and maternal safety.

The objective of our study is, therefore, to compare maternal and neonatal morbidity among women with a singleton fetus in a breech presentation at term by planned modes of labor and, specifically, to compare cervical ripening to the other modes of labor (IOL with a favorable cervix, spontaneous labor, or elective cesarean delivery before labor).

## 2. Materials and Methods

A retrospective review of prospectively collected data on consecutive women who gave birth to fetuses in a breech presentation between 1 January 2016 and 31 December 2020 in a tertiary care hospital with more than 2700 annual deliveries was carried out. For this study, we included consecutive pregnant women with a live singleton fetus in a breech presentation at term (>37 weeks of gestation), whatever the planned mode of labor (spontaneous labor, cervical ripening, IOL with favorable cervix, or elective cesarean delivery before labor). We excluded women with a preterm birth (<37 weeks), women without a first-trimester ultrasound gestational age dating (crown-to-rump length at a first-trimester ultrasound examination or fetal biometry before 24 weeks), multiple pregnancies, women with a medically indicated second-trimester termination of pregnancy, and intrauterine death or fetal loss before 22 weeks.

This present study was conducted in accordance with the French-approved guidelines. All participants received oral information about the study. Written consent was not required for the retrospective study according to French law, but each woman had the opportunity to opt-out of the analysis. The study protocol was approved by a Research Ethics Committee (*Groupe Nantais d’Ethique dans le Domaine de la Santé* (GNED), n°2021-06-24) on 24 June 2021, before the beginning of the study.

Four groups of included women at term with a breech presentation constituted the planned mode of labor (spontaneous labor, cervical ripening, IOL with favorable cervix, or elective cesarean delivery before labor).

Indications of the IOL were similar to those with a cephalic presentation but were also in accordance with French guidelines for planned vaginal delivery in a breech presentation at term [7]. They consisted of prolonged pregnancy (over 41 weeks + 4 days), prelabor rupture of the membranes, pregnancy-associated hypertensive disorders [8], fetal growth restriction (small-for-gestational-age, SGA, defined as an ultrasonographic estimated fetal weight < 10th centile, adjusting for gestational age and sex [9]), abnormality of fetal vitality (i.e., oligoamnios or decreased fetal movements before 41 weeks of gestation), diabetes (gestational [10] or previous; insulin- or noninsulin-dependent), macrosomia without diabetes (defined as an ultrasonographic estimated fetal weight > 90th centile adjusting for gestational age and sex [9]), and other medical indication (i.e., thrombocytopenia, intrahepatic cholestasis of pregnancy, hydramnios, and other maternal complications).

Pregnant women undergoing an IOL at 37 weeks of gestation or more with a Bishop score higher than 6 underwent their IOL with oxytocin infusion and/or amniotomy. Women with a Bishop score of less than 6 had preinduction cervical ripening using either 10 mg slow-release vaginal dinoprostone insert (Propess^®^, Ferring, Saint-Prex, Switzerland) or CRB (Cervical Ripening Balloon^®^, Cook OB/GYN, Spencer, IN, USA). The choice was at the physician’s discretion in accordance with the women’s choice after explications of each method and in accordance with the local institutional guidelines for cervical ripening. After cervical ripening, the women underwent fetal heart rate (FHR) monitoring immediately after insertion and every 6 h afterward. Dinoprostone vaginal insert and CRB are approved for use for up to 24 h. If labor did not ensue or the Bishop score was still unfavorable (score < 6) after 24 h, there was no consensus, and repeating the vaginal dinoprostone insert or switching to another method were options at the free discretion of the treating physician. The choice of the maximum duration of cervical ripening permitted before starting oxytocin if the membranes were not accessible was also left to the free discretion of the obstetrician responsible for the woman with a maximum of 3 days in our obstetric team. Once the Bishop score was ≥6, further management with oxytocin infusion and/or amniotomy was recommended by our team.

Women with a breech presentation at term and who elected to have a cesarean delivery before labor according to the usual indications (inadequate maternal pelvimetry, maternal request, estimated fetal weight (EFW) above 4000 g, and other obstetric indications (i.e., severe preeclampsia, severe thrombocytopenia...)) [7] were included in the fourth group.

The maternal characteristics that were collected included maternal age, prepregnancy body mass index (BMI, calculated as weight (kg)/[height (m)]^2^, based on height and the first weight noted in the obstetric record), women’s medical history (i.e., known uterine malformation, previous diabetes, previous hypertension, one previous cesarean delivery, etc.), parity, and type of conception (spontaneous or using assisted reproductive technology). 

Pregnancy and labor characteristics were collected regarding the mode of labor. For elective cesarean deliveries, indications, gestational age at delivery, and type of anesthesia were collected. For planned vaginal deliveries, the variables collected were information about indication of IOL, method and cervical status (Bishop score) before and after cervical ripening, uterine tachysystole, artificial rupture of membranes before starting oxytocin, labor (oxytocin induction, quantity of oxytocin used), time-interval from the IOL beginning to delivery, gestational age at delivery, mode of delivery (cesarean delivery before labor, spontaneous or operative vaginal delivery, and cesarean section during labor), indication of cesarean delivery, time of cesarean delivery during labor (during the latent or active phase), and birth weight. 

The maternal characteristics collected in the immediate postpartum period were severity of perineal tears, perineal hematoma, total estimated blood loss routinely assessed with a collector bag placed just after birth, postpartum hemorrhage (PPH, defined as bleeding 500 mL or greater) and severe PPH (defined as bleeding 1000 mL or greater), need for additional uterotonic agent (sulprostone) and second-line therapies (Bakri balloon, uterine compression sutures, uterine artery embolization, and peripartum hysterectomy) for the management of massive persistent PPH after failure of uterine massage and uterotonic agents to stop any bleeding, chorioamnionitis, or infections (defined by at least one of the following: endometritis, episiotomy infection, or wound infection requiring surgery), blood transfusion, thromboembolic event, intensive care unit (ICU) admission, and maternal death.

The neonatal outcomes collected included birth weight, fetal sex, umbilical artery blood gas values that were routinely measured, immediate neonatal data that were recorded with a systematic pediatrician examination after delivery (5 min Apgar score, neonatal trauma, need for resuscitation or intubation, and respiratory distress syndrome), neonatal hyperbilirubinemia that needed phototherapy after birth, neonatal sepsis (defined as confirmed clinical infection with positive bacteriological tests), admission to the neonatal intensive care unit (NICU), neonatal trauma, and neonatal death. Neonatal trauma was defined by at least one of the following criteria: subdural hematoma, spinal cord injury, basal skull fracture, fracture of a long bone or of the clavicle, peripheral nerve injury, and genital injury.

The endpoints were the composite variables of severe maternal and neonatal morbidity. The primary endpoint was severe short-term neonatal morbidity defined by at least one of the following criteria: 5 min Apgar score less than 7, pH less than 7.00, need for resuscitation or intubation, an NICU admission longer than 24 h, hyperbilirubinemia, sepsis, seizures, intraventricular hemorrhage greater than grade 2, neonatal trauma, and neonatal death. The secondary endpoint was severe maternal morbidity, defined by at least one of the following criteria: perineal hematoma, chorioamnionitis, third- or fourth-degree perineal tears, a PPH greater than 1000 mL, the need for additional uterotonic agents, second-line therapies, blood transfusion, infection, thromboembolic events, admission to the intensive care unit, and maternal death.

Continuous data were described by their means ± standard deviations and compared by *t*-tests (or Mann–Whitney tests when appropriate), and the categorical data were described by percentages and compared by χ^2^ tests (or Fisher’s exact tests when appropriate). We compared the maternal and neonatal outcomes according to the planned mode of labor (spontaneous labor, IOL with favorable cervix, cervical ripening, and planned cesarean delivery before labor), and specifically studied the association (assessed by multivariate logistic regression analyses) between maternal and neonatal morbidity and cervical ripening (compared with planned cesarean delivery). The multivariate logistic regression allowed us to analyze (together) the effect of other risk factors and potential confounders (maternal age, parity, BMI before pregnancy, gestational age at delivery, mode of labor, mode of delivery, and birth weight) [3]. We used R software (version 4.1.2) for all the analyses. *p* values < 0.05 were considered to be statistically significant.

## 3. Results

During the study period, 12,843 births took place in our tertiary public hospital, and 362 women (2.8%) with a live singleton fetus in a breech presentation at term were managed in our hospital. Among these women, 178 (49.1%) had a planned cesarean delivery before labor, and 184 (50.8%) had a planned vaginal delivery, including 143 of them (39.5%) who had spontaneous labor, 14 (3.9%) who underwent the IOL with a favorable cervix and 27 (7.5%) who had cervical ripening due to an unfavorable cervix (Figure 1).

The maternal characteristics according to the type of labor are shown in Table 1. The mean gestational age at delivery was 39.3 ± 1.1 weeks of gestation and was similar between the groups (*p* = 0.17). More previous cesarean deliveries were observed among women with an elective cesarean delivery compared to the others groups (Table 1). Among the 178 women who had an elective cesarean delivery, various indications were observed: inadequate maternal pelvimetry (53/178, 29.8%), maternal request (62/178, 34.8%), EFW above 4000 g (32/178, 18.0%), and other obstetric indication (i.e., severe preeclampsia, severe thrombocytopenia, etc.) (31/178, 17.4%). Neuraxial analgesia was almost always used for elective cesarean deliveries (171/178, 96.1%), and seven general anesthesias were required due to contraindications to neuraxial analgesia. Among the women who underwent a planned vaginal delivery (184/362, 50.8%), vaginal delivery was significantly less frequent after cervical ripening (63.0%) compared to spontaneous labor (88.8%) or the IOL with a favorable cervix (92.9%) (*p* = 0.01) (Table 1).

Prolonged pregnancy was the most frequent indication for the IOL (15/41, 36.6%), and the prelabor rupture of membranes was significantly more frequent among women undergoing an IOL and a favorable cervix (50.0% vs. 3.7%, *p* = 0.001) (Table 2). 

For cervical ripening, a CRB was the most frequently used device (20/27, 74.1%), and four women (4/27, 14.8%) required repeated dinoprostone vaginal inserts.

The severe short-term neonatal morbidity among women who gave birth to a live singleton fetus in a breech presentation at term was 3.0% (11/362), and this was significantly higher after the IOL with a favorable cervix (14.2%) or cervical ripening (7.4%) compared to elective cesarean delivery (1.2%) or spontaneous labor (3.5%) (*p* = 0.02) (Table 3 and Table 4).

Women with severe neonatal morbidity differed according to prepregnancy BMI but not according to the method of cervical ripening (50% after CRB vs. 50% after dinoprostone, *p* = 0.52). After adjusting for the confounding factors in the multivariate logistic regression analysis, the risk of severe, composite short-term neonatal morbidity was not significantly different among cervical ripening and elective cesarean delivery (adjusted odds ratio [OR] 2.80, 95% confidence interval [CI] 0.10–43.6) (Table 5).

Severe maternal morbidity among women who gave birth to a live singleton fetus in a breech presentation at term was 6.3% (23/362). Severe maternal morbidity rates were likewise similar (4.9% after spontaneous labor, 0% after the IOL with a favorable cervix, 7.4% after cervical ripening, and 7.9% after elective cesarean delivery before labor; *p* = 0.70) (Table 1). After controlling for risk factors, cervical ripening (compared to elective cesarean delivery) was not significantly associated with severe maternal morbidity (adjusted OR 1.29, 95% CI 0.05–11.5) (Table 5).

## 4. Discussion

In our study, cervical ripening among women with a breech presentation at term was not associated with a high rate of severe maternal and neonatal morbidity when compared to elective cesarean delivery. In the univariate analysis, composite severe neonatal morbidity was less frequent in the planned cesarean delivery group, but this difference did not persist after multivariate analysis.

Our reported severe short-term neonatal morbidity (3.0%) is consistent with other well-established findings in the literature [3]. Concerning severe neonatal morbidity after the IOL, our rate (9.8%) is higher than those reported in the literature (1–2%) [11,12]. Nonetheless, the rates of NICU admission (3.7%) and a 5 min Apgar score of <7 (3.7%) after cervical ripening were consistent with other studies [6,13,14,15]. Furthermore, we would like to underline that our criteria for neonatal morbidity were less severe than those used in the secondary analysis of the PREMODA study (a 5 min Apgar score of less than 4, intubation persistent after the first 24 h, NICU admission longer than four days hours, convulsions continued after the first 24 h, neonatal trauma, and neonatal death) [12] or in a large retrospective comparative hospital-based study including 597 women (96 women who had labor induced and 501 in spontaneous labor) [11].

The CRB was the most frequently used device for cervical ripening (20/27, 74.1%). This rate was higher than in a large retrospective cohort study of women with singleton live fetuses in a breech presentation who required cervical ripening (55.1%) [16]. Few data are reported for a comparison of the different methods of cervical ripening among women with a breech presentation at term [17,18], and the use of CRBs and dinoprostone seemed to be similar for neonatal morbidity. Our results were restricted by the limited number of severe maternal (*n* = 1) and neonatal (*n* = 1) complications in our sample after CRB use (*n* = 20) or dinoprostone insert (*n* = 7) as the first method used for cervical ripening, from which we are unable to study the impact of the method on perinatal outcomes.

In our study, severe maternal morbidity (6.3%) was higher than previously reported figures (2–3%) [11,12,19] but was not different according to the planned mode of labor, and cervical ripening (compared to elective cesarean delivery) was not significantly associated with severe maternal morbidity. In the largest study among the 7564 women included in the secondary analysis of the PREMODA study, the short-term risk of severe maternal morbidity was 0.9% and did not differ significantly according to the planned mode of delivery (0.9% after planned cesarean delivery vs. 0.7% after a planned vaginal delivery, with an adjusted risk ratio of 1.60, 95% confidence interval 0.81–3.15) [19].

The principal strength of our study is that all the included pregnant women with a live singleton fetus in a breech presentation at term in our center, no matter the planned mode of labor, were managed by the same obstetric team throughout the study period, which avoided significant variations regarding the clinical management of a breech presentation at term, including the validation of an attempted vaginal delivery, the acceptance of an IOL, and cervical ripening. We reported a rate of labor induction among breech presentations with a planned vaginal delivery (41/184, 22.2%) that is consistent with the French rate of labor induction, no matter the fetal presentation [1]. Second, we reported a higher rate of cervical ripening with CRB use (74.1%) than a large retrospective cohort study of women with singleton live fetuses in a breech presentation who required cervical ripening (55.1%) [16], and CRB use seemed to be more efficient and as safe compared to prostaglandins for cervical ripening among women with a breech presentation at term in the few reported data in the literature [17,18]. Our results must be interpreted in light of certain limitations. First, our study reflects the experience of one tertiary hospital, and its results can be generalized only to other perinatal centers using the same practices (skilled obstetricians, daily discussions, and the validation of attempted vaginal deliveries in women with a breech presentation at term). Second, we reported a small sample size (*n* = 27) and a lower rate of cervical ripening (7.5%) compared to another French study (31.4%) [12]. This, too, may limit the generalizability of our results, which may not impact clinical practice. Third, although the sample size of this retrospective review of a prospective cohort was large (*n* = 362), including a substantial number of IOLs (*n* = 41, 11.3%), and is comparable in size to other published studies on maternal or neonatal outcomes after the IOL in women with a breech presentation at term [11,20], the limited number of severe maternal (*n* = 23) and neonatal (*n* = 11) complications in our sample might not have been high enough to reveal the statistically significant effects of the intervention. Furthermore, the limited number of severe maternal (*n* = 1) and neonatal (*n* = 1) complications in our sample after CRB use (*n* = 20) or dinoprostone insert (*n* = 7) as the first method used for cervical ripening has limited the comparison of our perinatal outcomes according to the method of cervical ripening.

## 5. Conclusions

The limitations notwithstanding, our study supports the continued use of a policy of offering cervical ripening to the appropriately selected candidates with a singleton fetus in a breech presentation at term. Indeed, cervical ripening, when compared to elective cesarean delivery, seemed to not be associated with an increase in severe short-term maternal or neonatal morbidity.

## Figures and Tables

**Figure 1 jcm-11-07118-f001:**
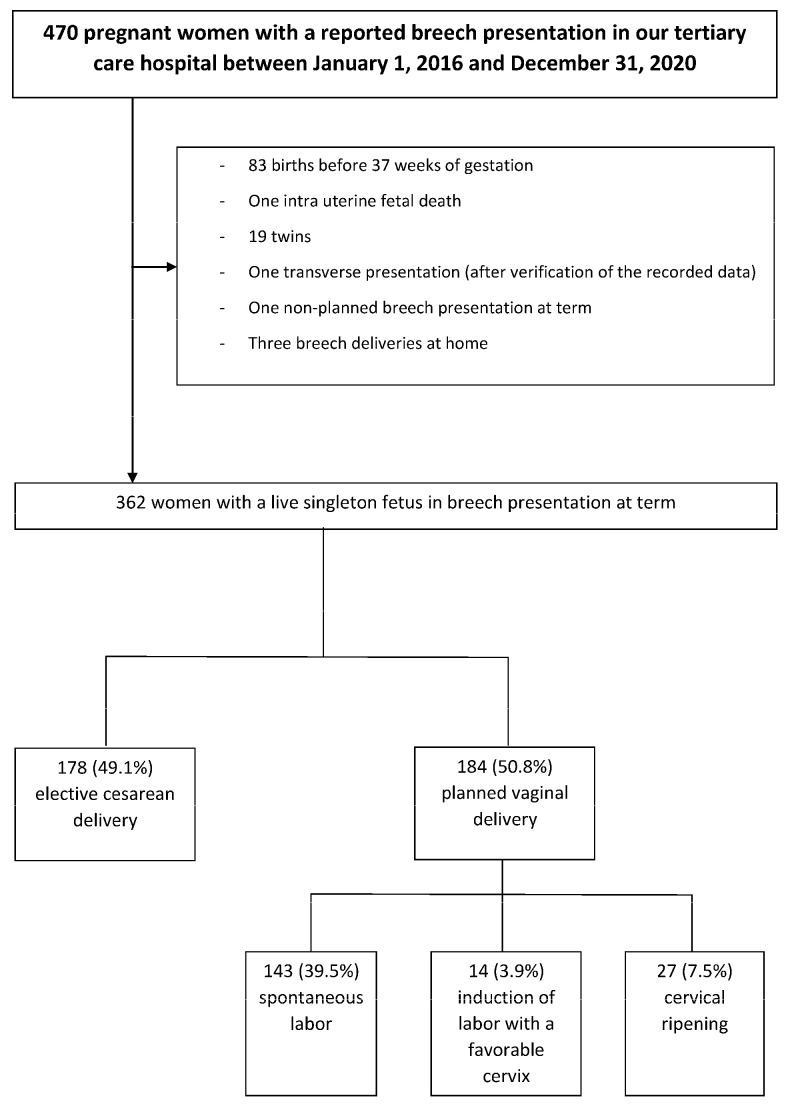
Flow-chart.

**Table 1 jcm-11-07118-t001:** Maternal and labor characteristics, and maternal outcomes according to the type of labor.

	Planned Cesarean Delivery, *n* = 178 (49.1%)	Spontaneous Labor, *n* = 143 (39.5%)	IOL with Favorable Cervix, *n* = 14 (3.9%)	Cervical Ripening, *n* = 27 (7.5%)	*p* Value
Age, years	30.5 ± 4.7	30.0 ± 4.2	29.4 ± 4.3	30.6 ± 3.9	0.80
Pre-pregnancy BMI, kg/m^2^	23.9 ± 4.8	23.1 ± 5.1	24.2 ± 6.3	25.2 ± 5.3	0.07
Obesity (BMI ≥ 30 kg/m^2^)	21 (11.8)	16 (11.2)	2 (14.2)	4 (14.8)	0.87
Preexisting type 1 or 2 diabetes	1 (0.6)	0	0	0	0.95
Chronic hypertension	3 (1.7)	0	0	1 (3.7)	0.20
Nulliparity	93 (52.2)	75 (52.4)	9 (64.3)	15 (55.6)	0.80
Uterine malformation	7 (3.9)	8 (5.6)	1 (7.1)	2 (7.4)	0.50
One previous cesarean delivery	40 (22.5)	6 (4.2)	0	1 (3.7)	<0.001
ART	15 (8.4)	8 (5.6)	1 (7.1)	3 (11.1)	0.34
Gestational age at delivery, weeks	39.0 ± 0.8	39.5 ± 1.1	40.0 ± 1.4	40.1 ± 1.6	0.17
Elective cesarean delivery before labor	-	-	0	4 (14.8)	0.64
Mode of delivery					
Spontaneous vaginal delivery	-	124 (86.7)	13 (92.9)	17 (63.0)	0.01
Operative vaginal delivery	-	3 (2.1)	0	0	0.90
Cesarean delivery during labor	-	16 (11.2)	1 (7.1)	6 (22.2)	0.16
Indication for cesarean delivery during labor					
Non-reassuring FHR only	-	9 (56.3)	0	5 (83.3)	0.04
Arrested progress only	-	5 (31.2)	1 (100)	1 (16.7)	0.02
Non-reassuring FHR and arrested progress	-	2 (12.5)	0	0	0.72
Postpartum hemorrhage	26 (14.6)	12 (8.4)	1 (7.1)	4 (14.8)	0.30
Severe PPH	5 (2.8)	3 (2.1)	0	0	0.96
Episiotomy	-	31 (21.7)	3 (21.4)	4 (14.8)	0.17
Third- or fourth-degree perineal	-	2 (1.4)	0	0	0.72
Perineal hematoma	-	0	0	1 (3.7)	0.11
Need for additional uterotonic agent (sulprostone)	1 (0.6)	0	0	1 (3.7)	0.26
Second-line therapies	2 (1.1)	0	0	0	0.54
Chorioamnionitis	0	2 (1.4)	1 (7.1)	0	0.05
Infections	4 (2.2)	4 (2.8)	0	0	0.51
Blood transfusion	2 (1.1)	2 (1.4)	0	0	0.92
Thromboembolic event	0	0	0	0	-
Intensive care unit admission	2 (1.1)	0	0	0	0.63
Maternal death	0	0	0	0	-
Maternal morbidity	14 (7.9)	7 (4.9)	0	2 (7.4)	0.70

Values are given as mean ± SD or number (percentage) unless otherwise indicated. *BMI*, body mass index, *ART*, assisted reproductive technology, *FHR*, fetal heart rate, *PPH*, postpartum hemorrhage. Severe PPH was defined as bleeding 1000 mL or greater. Second-line therapies were defined as Bakri balloon, uterine compression sutures, uterine artery embolization, and peripartum hysterectomy for the management of massive persistent. PPH after failure of uterine massage and uterotonic agents to stop the bleeding. Infections were defined by at least one of the following: endometritis, episiotomy infection, or wound infection requiring surgery. Maternal morbidity was defined by at least one of the following criteria: perineal hematoma, chorioamnionitis, third- or fourth-degree perineal, a PPH greater than 1000 mL, the need for an additional uterotonic agent, second-line therapies, blood transfusion, infection, thromboembolic events, admission to the intensive care unit, and maternal death.

**Table 2 jcm-11-07118-t002:** Maternal and induction of labor characteristics.

	IOL with Favorable Cervix, *n* = 14 (4.0%)	Cervical Ripening, *n* = 27 (7.5%)	*p* Value
Gestational age at IOL, weeks	40.0 ± 1.4	40.1 ± 1.6	0.63
Indication for IOL			
Prolonged pregnancy	4 (28.7)	11 (40.8)	0.47
Diabetes	0	3 (11.1)	0.87
SGA	1 (7.1)	4 (14.8)	0.54
Pregnancy-associated hypertensive disorders	0	1 (3.7)	0.52
Prelabor rupture of membranes	7 (50.0)	1 (3.7)	0.001
Intrahepatic cholestasis of pregnancy	0	1 (3.7)	0.52
Abnormality of fetal vitality	1 (7.1)	4 (14.8)	0.54
Other medical indication	1 (7.1)	2 (7.4)	0.99
Bishop score before IOL	6.9 ± 1.4	3.4 ± 0.4	<0.0001
Methods of cervical ripening			
CRB alone	-	20 (74.1)	-
Dinoprostone vaginal insert alone	-	3 (11.1)	-
Repeated dinoprostone vaginal insert	-	4 (14.8)	-
Time-interval from IOL beginning to delivery, hours	9.0 ± 3.0	24.0 ± 14.0	<0.0001

Values are given as mean ± SD or number (percentage) unless otherwise indicated. *IOL*, induction of labor, SGA, small-for-gestational age, *CRB*, cervical ripening balloon.

**Table 3 jcm-11-07118-t003:** Neonatal outcomes according to the type of labor.

	Planned Cesarean Delivery, *n* = 178 (49.2%)	Spontaneous Labor, *n* = 143 (39.5%)	IOL with Favorable Cervix, *n* = 14 (4.0%)	Cervical Ripening, *n* = 27 (7.5%)	*p* Value
Male	81 (45.6)	61 (42.7)	4 (28.6)	8 (29.6)	0.30
Birth weight, g	3215 ± 505	3098 ± 365	3176 ± 399	3129 ± 511	0.06
5 min Apgar score less than 7	0	2 (1.4)	1 (7.1)	1 (3.7)	0.02
pH less than 7.00	0	4 (2.8)	0	2 (7.4)	0.02
Need for resuscitation or intubation	1 (0.6)	1 (0.7)	0	1 (3.7)	0.41
NICU admission longer than 24 h	2 (1.2)	2 (1.4)	2 (14.2)	1 (3.7)	0.03
Respiratory distress syndrome	2 (1.2)	2 (1.4)	0	2 (7.4)	0.20
Neonatal hyperbilirubinemia	1 (0.6)	0	1 (7.1)	0	0.08
Sepsis	0	0	0	0	-
Seizures	0	0	0	1 (3.7)	0.11
Intraventricular hemorrhage greater than grade 2	0	0	0	0	-
Neonatal trauma	0	0	0	0	-
Neonatal death	0	0	0	1 (3.7)	0.11
Severe neonatal morbidity	2 (1.2)	5 (3.5)	2 (14.2)	2 (7.4)	0.02

Values are given as mean ± SD or number (percentage) unless otherwise indicated. *NICU*, neonatal intensive care unit. Neonatal trauma was defined as by at least one of the following criteria: subdural hematoma, spinal cord injury, basal skull fracture, fracture of a long bone or of the clavicle, peripheral nerve injury, and genital injury. Severe neonatal morbidity was defined by at least one of the following criteria: a 5 min Apgar score less than 7, a pH less than 7.00, the need for resuscitation or intubation, NICU admission longer than 24 h, neonatal hyperbilirubinemia, neonatal sepsis, seizures, an intraventricular hemorrhage greater than grade 2, neonatal trauma, and neonatal death.

**Table 4 jcm-11-07118-t004:** Univariate analysis of severe maternal and neonatal morbidity after planned cesarean delivery or spontaneous or induced labor.

	Severe Maternal Morbidity *			Severe Neonatal Morbidity **		
Variable	No (*n* = 339)	Yes (*n* = 23)	*p* Value	No (*n* = 351)	Yes (*n* = 11)	*p* Value
Age, years	30.3 ± 4.3	29.7 ± 6.3	0.40	30.3 ± 4.5	28.3 ± 2.6	0.06
Pre-pregnancy BMI, kg/m^2^	23.7 ± 5.1	23.6 ± 3.2	0.40	23.8 ± 5.0	20.9 ± 3.3	0.04
Obesity (BMI ≥ 30 kg/m^2^)	43 (12.7)	0	0.09	43 (12.2)	0	0.42
Nulliparity	176 (51.9)	16 (69.6)	0.10	184 (52.4)	8 (72.7)	0.23
Gestational age at delivery, weeks	39.3 ± 1.1	39.2 ± 1.1	0.44	39.3 ± 1.1	39.9 ± 1.3	0.22
Mode of labor			0.77			0.02
Elective cesarean delivery	164 (48.4)	14 (60.9)		176 (50.1)	2 (18.2)	
Spontaneous labor	136 (40.1)	7 (30.4)		138 (39.3)	5 (45.4)	
IOL with favorable cervix	14 (4.1)	0		12 (3.4)	2 (18.2)	
Cervical ripening	25 (7.4)	2 (8.7)		25 (7.2)	2 (18.2)	
Method of cervical ripening			0.52			0.52
Cervical ripening balloon	19 (76.0)	1 (50.0)		19 (76.0)	1 (50.0)	
Dinoprostone vaginal insert	5 (20.0)	1 (50.0)		5 (20.0)	1 (50.0)	
Repeated dinoprostone insert	1 (4.0)	0		1 (4.0)	0	
Cesarean delivery	190 (56.0)	15 (65.2)	0.40	202 (57.6)	3 (27.3)	0.06
Birth weight, g	3158 ± 445	3195 ± 572	0.81	3159 ± 453	3215 ± 466	0.72

Values are given as mean ± SD or number (percentage) unless otherwise indicated. *IOL*, induction of labor. * Severe maternal morbidity was defined by at least one of the following criteria: perineal hematoma, chorioamnionitis, third- or fourth-degree perineal, a PPH greater than 1000 mL, the need for an additional uterotonic agent, second-line therapies, blood transfusion, infection, thromboembolic events, admission to the intensive care unit, and maternal death. ** Severe neonatal morbidity was defined by at least one of the following criteria: a 5 min Apgar score less than 7, a pH less than 7.00, the need for resuscitation or intubation, NICU admission longer than 24 h, neonatal hyperbilirubinemia, neonatal sepsis, seizures, intraventricular hemorrhage greater than grade 2, neonatal trauma, and neonatal death.

**Table 5 jcm-11-07118-t005:** Multivariate analysis of severe maternal and neonatal morbidity after planned cesarean delivery or spontaneous or induced labor.

	Severe Maternal Morbidity (*n* = 23)		Severe Neonatal Morbidity (*n* = 11)	
Variable *	Adjusted OR (95% CI)	*p* Value	Adjusted OR (95% CI)	*p* Value
Age (/year)	1.00 (0.90–1.10)	0.93	0.93 (0.79–1.09)	0.41
Nulliparity	2.01 (0.72–6.57)	0.21	1.25 (0.29–6.57)	0.84
Multiparity	Reference	-	Reference	-
Gestational age at delivery, weeks				
Less than 39 weeks	1.16 (0.46–2.84)	0.77	1.28 (0.24–6.08)	0.81
39 to less than 41	Reference	-	Reference	-
Greater than 41	0.26 (0.01–1.67)	0.22	2.76 (0.51–14.2)	0.24
Mode of labor				
Elective cesarean delivery	Reference	-	Reference	-
Spontaneous labor	0.52 (0.03–3.45)	0.61	1.36 (0.05–19.2)	0.84
IOL with favorable cervix	0	-	4.60 (0.11–106)	0.42
Cervical ripening	1.29 (0.05–11.5)	0.81	2.80 (0.10–43.6)	0.53
Mode of delivery				
Vaginal delivery	Reference	-	Reference	-
Cesarean delivery during labor	0.82 (0.04–5.45)	0.94	0.54 (0.02–4.13)	0.62

OR, odds ratio; CI, confidence interval; BMI, body mass index; IOL, induction of labor. * Adjusted for maternal age, parity, BMI before pregnancy, gestational age at delivery, mode of labor, mode of delivery, and birth weight.

## Data Availability

The data presented in this study are available on request from the corresponding author.

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
