# Peer review of "Neonatal Morbidity after Cervical Ripening with a Singleton Fetus in a Breech Presentation at Term"

_jcm, 2022, doi:10.3390/jcm11237118_

Round 1

Reviewer 1 Report

167: Can we say that spontaneous labor is a planned vaginal delivery?

Table 1: Can we say that patients who were induced had spontaneous vaginal delivery?

Regarding the well mentioned limitations of the study, the conclusions sound too strong and too definite. Being aware of the low prevalence of breech presentation at term managed with cervical ripening and with a great value of data shared by the authors, we still need to understand the low statistical relevance of this study results. Comparing the outcome of subgroups with hundreds and 14 or 27 patients cannot impact the clinical practice.

Reviewer 2 Report

The authors examined the neonatal morbidity after cervical ripening with a singleton fetus in breech presentation at term. The concept of this study is of some interest. However, this study has severe flaws regarding the methodology. The number of included women was limited, and multivariable logistic regression analysis was inappropriate due to the limited number of eligible cases.

Reviewer 3 Report

It is an interesting work that shows the institutional experience concerning vaginal birth in fetuses with breech presentation. However, this subject continues to be controversial due to the idea that the risks for the mother or the fetus are increased. 

The experience in their institution has some drawbacks; for example, the groups are unbalanced, the elective cesarean group has 178 patients vs. the IOL group with favorable cervix 14. 

The statistical analysis described in the methods mentions tests to compare two groups and, on some occasions, compares the four groups in which they classify their patients. It is suggested that the statistical tests used to be corrected. 

The multivariate analysis is an outstanding contribution to helping understand the results. However, it is suggested to add variables such as fetal weight since this could be the explanation for the results that have little biological support in the bivariate analysis. 

It is simple work, but it helps to demonstrate a controversial point in obstetric practice. 

It is well written, and the results are well presented; without being a pretentious work, it contributes to the field of knowledge.

Round 2

Reviewer 2 Report

The authors revised the manuscript according to the reviewers' comments. While I have concerns regarding the statistical method, the authors have not addressed this point. To achieve enough statistical power, a minimum of 20 observations of the outcome per variable should be included in the model.